# Automated virtual reality (VR) cognitive therapy for patients with psychosis: study protocol for a single-blind parallel group randomised controlled trial (gameChange)

Daniel Freeman ![ORCID],[1,2,3] Ly-Mee Yu,[4] Thomas Kabir,[5] Jen Martin,[6,7,8] Michael Craven,[6,8,9] José Leal,[10] Sinéad Lambe,[1,2,3] Susan Brown,[6,7,8] Anthony Morrison,[11,12] Kate Chapman,[13] Robert Dudley,[14,15] Eileen O'Regan,[16] Aitor Rovira,[1,2,3] Andrew Goodsell,[1,2,3] Laina Rosebrock,[1,2,3] Aislinn Bergin,[6,7,8] Tillie L Cryer,[5] Dan Robotham,[5] Humma Andleeb,[5] John R Geddes,[1,2,3] Chris Hollis,[6,7,8] David M Clark,[2,3,17] Felicity Waite ![ORCID] [1,2,3]

For numbered affiliations see end of article.

**Correspondence to**
Daniel Freeman;
daniel.freeman@psych.ox.ac.uk

## ABSTRACT

**Introduction** Many patients with psychosis experience everyday social situations as anxiety-provoking. The fears can arise, for example, from paranoia, hallucinations, social anxiety or negative-self beliefs. The fears lead patients to withdraw from activities, and this isolation leads to a cycle of worsening physical and mental health. Breaking this cycle requires highly active treatment directly in the troubling situations so that patients learn that they can safely and confidently enter them. However patients with psychosis seldom receive such life-changing interventions. To solve this problem we have developed an automated psychological treatment delivered in virtual reality (VR). It allows patients to experience computer simulations of the situations that they find anxiety-provoking. A virtual coach guides patients, using cognitive techniques, in how to overcome their fears. Patients are willing to enter VR simulations of anxiety-provoking situations because they know the simulations are not real, but the learning made transfers to the real world.

**Methods and analysis** 432 patients with psychosis and anxious avoidance of social situations will be recruited from National Health Service (NHS) secondary care services. In the gameChange trial, they will be randomised (1:1) to the six-session VR cognitive treatment added to treatment as usual or treatment as usual alone. Assessments will be conducted at 0, 6 (post-treatment) and 26 weeks by a researcher blind to allocation. The primary outcome is avoidance and distress in real-life situations, using a behavioural assessment task, at 6 weeks. The secondary outcomes are psychiatric symptoms, activity levels and quality of life. All main analyses will be intention-to-treat. Moderation and mediation will be tested. An economic evaluation will be conducted.

**Ethics and dissemination** The trial has received ethical approval from the NHS South Central - Oxford B Research Ethics Committee (19/SC/0075). A key output will be a high-quality automated VR treatment for patients to overcome anxious avoidance of social situations.

## Strengths and limitations of this study

► A multicentre randomised controlled trial of 432 patients with psychosis being seen in National Health Service mental health trusts, which will be the largest trial of virtual reality (VR) used to treat a mental health condition.

► Automated delivery of the VR intervention meaning high treatment fidelity and a highly scalable treatment that could greatly increase access to psychological therapy.

► Mediation built into the treatment design can test whether the treatment works as hypothesised.

► The control condition is treatment as usual meaning that it cannot be definitively established which VR treatment elements produce clinical change.

► It is impossible to blind patients to the treatment allocation, which could introduce bias into the treatment effect estimation.

**Trial registration number** ISRCTN17308399.

## BACKGROUND

### Rationale

Too many patients with psychosis, despite standard treatment, become isolated and inactive, with negative effects on both mental and physical health. Approximately 80% of patients with schizophrenia experience an episode of depression.[1] Physical activity levels in patients with schizophrenia are reduced on average by approximately two-thirds.[2] Over 90% of patients with schizophrenia are unemployed and spend 'less time in functional but also in social and leisure activities and more time resting and 'doing nothing'

compared with the general population'.[3] Life expectancy is on average 14.5 years shorter,[4] due to largely preventable conditions such as high blood pressure, diabetes and heart disease. Partly this physical ill health reflects unhealthy lifestyles including inactivity.

Our view is that a substantial part of this inactivity arises from avoidance due to anxiety. In a clinical assessment study of 1800 patients with non-affective psychosis attending National Health Service mental health services, two-thirds of the patients had levels of anxious avoidance equivalent to patients diagnosed with agoraphobia.[5] The anxiety in patients with psychosis can arise from a number of sources: fears that others will harm them, voices telling them of danger, social anxiety fears of humiliation and rejection and negative beliefs about the self that cause a lack of confidence and a sense of vulnerability. But withdrawal from activities because of anxiety need not be inevitable. Appropriate treatment, as seen in the anxiety disorders,[6] can produce excellent outcomes. Such treatment involves identifying fearful thoughts and the safety-seeking (or defence) behaviours that maintain those cognitions by preventing receipt and processing of disconfirmatory evidence. The thoughts must then be tested in behavioural experiments in the troubling situations while the defence behaviours are dropped.[7] However, there is a dearth of therapists to carry out this skilled work for patients with schizophrenia. It is well-recognised that there is considerable under-provision of psychological therapy for patients with schizophrenia.[8] There is the additional problem that sometimes very fearful beliefs of patients with psychosis mean that they can be much less likely to engage in behavioural experiments in the real world before their fears have been lessened by other means. Our solution is the provision of automated psychological therapy using virtual reality (VR).

Virtual reality (interactive computer-generated environments) has been used since the early 1990's to treat anxiety.[9] Meta-analyses indicate that VR treatments for anxiety disorders can produce large treatment effects[10] that generalise to the real world.[11] Previous uses of VR for mental health problems have depended on a therapist providing the psychological therapy.[12] In a trial of 100 patients with a fear of heights, we have shown that the provision of cognitive therapy can be automated using VR by the incorporation of a virtual coach.[13] The treatment effect sizes in this trial were very large (effect size Cohen's d=2.0; the number of patients needed to treat to at least halve fear of heights was 1.3), and better than expected from face-to-face therapy. Automated treatment has the potential to be scalable, removing a key cause of the highly limited access to psychological therapy for patients with psychosis.

VR may also be especially suited to the difficulties of patients with psychosis. Patients with strong fears are much more likely to test out their fear expectations in VR because they know it is a simulation but the learning that they make then transfers to the real world. VR treatment can also include engaging tasks that make the treatment experience much more pleasurable. A graded approach can easily be applied in VR, allowing the individual to repeatedly experience the situations they find difficult and make new learning. Our view is that VR treatments have the potential to be faster, more efficacious and appealing to patients than traditional face-to-face approaches. We conducted a first test of VR to treat persecutory delusions in patients with psychosis.[14] Just 30 minutes in graded VR environments, with the psychological advice provided by a therapist, led to a large reduction in distress in real-world situations (eg, going into a shop). VR has been shown to be safe to use with patients with psychosis.[15] A recent randomised controlled trial of over 100 patients with psychosis showed that sixteen 1hour sessions with VR environments and a therapist who administered cognitive behavioural therapy techniques led at follow-up to a moderate increase in time spent with other people as assessed by an experience sampling method.[16] In the THerapeutic Realistic Immersive Virtual Environments (THRIVE) trial our team is currently testing a four-session automated VR cognitive treatment for patients specifically with persecutory delusions (ISRCTN12497310).[17]

In the gameChange project (www.gameChangeVR.com), we have recently developed - using a socially-inclusive design process - a new automated VR cognitive treatment for patients for psychosis having difficulties being in everyday social situations due to anxiety. It is designed to be easy to use, engaging for patients and staff and delivered with the latest consumer equipment. Therefore this VR treatment has the potential to be widely implemented in treatment services. Psychological treatment that involves direct coaching in the situations that trouble patients with psychosis is rarely available in mental health services. Therefore we set out to determine the *in toto* effects of adding the VR treatment to treatment as usual. This entails a test that randomises patients to receive the VR treatment in addition to usual care or to usual care. We aim to determine the clinical effects on real-world performance, activity levels, psychiatric symptoms and quality of life.

### Aims and hypotheses

The primary research question we aim to test is: Does automated VR cognitive treatment added to treatment as usual, compared with treatment as usual alone, lead to a post-treatment reduction in real world avoidance and distress for patients with psychosis attending NHS mental health services?

Our primary hypothesis is that:

1. Compared with treatment as usual, VR cognitive therapy added to treatment as usual will reduce avoidance and distress of real world situations (post-treatment).

Our secondary hypotheses are:

1. Compared with treatment as usual, VR cognitive therapy added to treatment as usual will reduce psychiatric symptoms (paranoia, anxious avoidance, depression,

suicidal ideation), increase activity and improve quality of life (post-treatment).

2. Treatment effects will be maintained at follow-up (6 months).
3. The mediators of VR treatment will be safety beliefs, threat cognitions and defence behaviours.
4. Treatment effects will be moderated by the occurrence of negative auditory hallucinations in social situations, hopelessness, appearance concerns and threat cognitions.

We also include a health economic evaluation of the VR treatment. It will focus on determining the cost of the VR treatment using a microcosting approach, performing a within-trial cost-effectiveness analysis and extrapolating the within-trial results to a 10 years horizon using a state-transition model.

## METHODS AND ANALYSIS
### Trial design and flow chart
The design is a multicentre, parallel group randomised controlled trial with single blind assessment to test whether the automated VR cognitive treatment added to treatment as usual, compared with treatment as usual alone,

leads to a post-treatment reduction in real world distress and avoidance for patients with psychosis attending NHS mental health services. Treatment as usual will be measured but remain unchanged in both groups. Assessments will be carried out at 0 (baseline), 6 (post-treatment) and 26 (follow-up) weeks by a researcher blind to treatment allocation. A summary of the trial design can be seen in figure 1. The trial is prospectively registered with the ISRCTN registry: ISRCTN17308399. There is a Data Monitoring and Ethics Committee.

### Randomisation, blinding and code-breaking
Participants will be randomised once they have completed the baseline assessment. Participants will be allocated to one of the trial arms using a 1:1 allocation ratio. Randomisation will be carried out by a validated online system, Sortition, designed by the University of Oxford Primary Care Clinical Trials Unit. Randomisation using a permuted blocks algorithm, with randomly varying block size, will be stratified by site (Bristol/Manchester/Newcastle/Nottingham/Oxford) and service type (in-patient/early intervention/community mental health team).

The research assessors will be blinded to group allocation, but the patients and staff member present will not

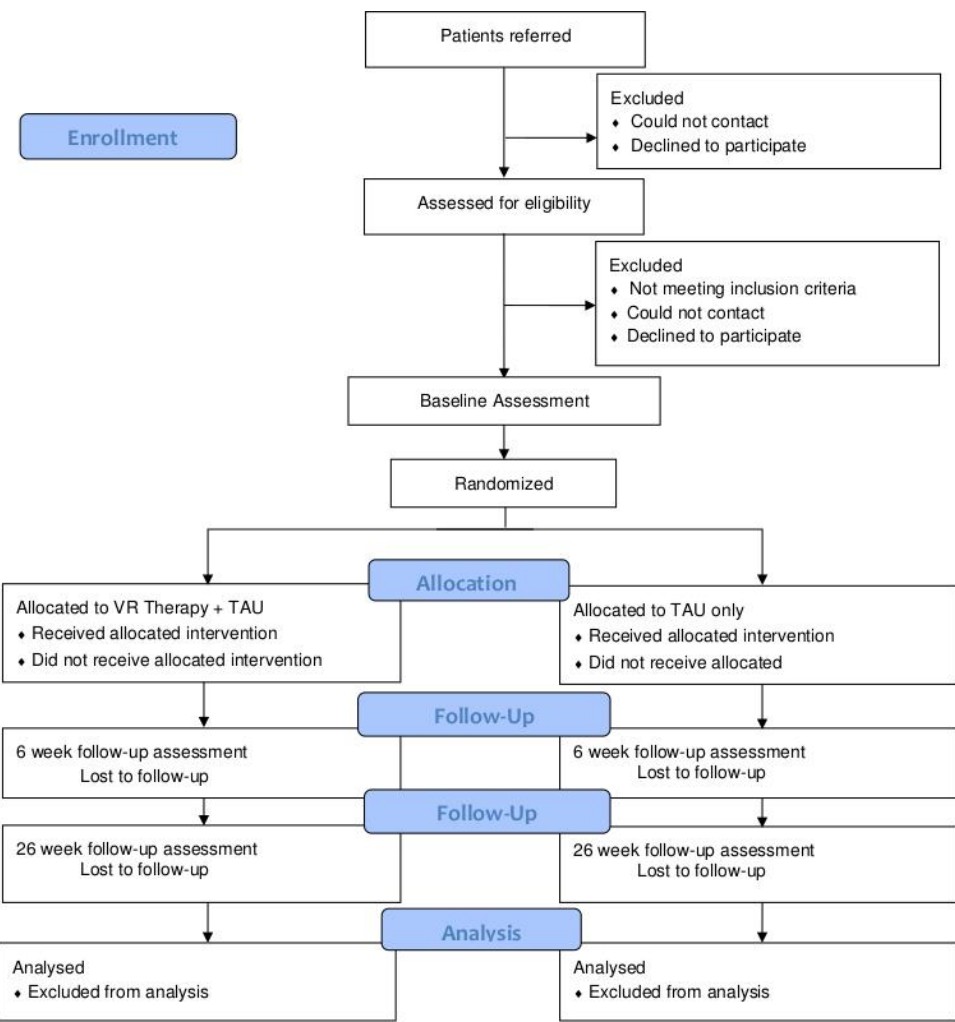

**Figure 1** Trial flow diagram. TAU, treatment as usual; VR, virtual reality.

be (they cannot be blinded to whether psychological intervention is delivered or received). If an allocation is revealed between assessment sessions, this is logged by the trial coordinator and re-blinding will occur using another assessor.

## Participants

The trial participants will be patients with psychosis and self-reported difficulties going outside among other people due to anxiety. The principal method of recruitment will be via seeking referrals to the trial from the relevant clinical teams (adult community mental health teams; early intervention services, and inpatient units) in the participating mental health trusts. The trial centres will be Bristol, Manchester, Newcastle, Nottingham, Oxford, with recruitment from local NHS mental health trusts. With the approval of the clinical team, patients interested in taking part will then be approached by the research team, given information about the trial, and screening conducted. Our Lived Experience Advisory Panel (LEAP) have also emphasised the importance of patients of the participating trusts self-initiating referral to the trials, in order to minimise the chances that particular patients are overlooked by clinical teams or the clinician was not present at a referral meeting. Hence we will also advertise the study and patients within participating trusts will be able to self-refer for a screening to take part in the study. However, in all instances we will also seek to confirm that a member of the clinical team gives approval for a patient to enter the trial. Informed consent will be obtained from all patients before participation.

## Inclusion criteria

► Adults aged 16 years or older;
► Attending an NHS mental health trust for the treatment of psychosis;
► Clinical diagnosis of schizophrenia spectrum psychosis (F20 to F29) or an affective diagnosis with psychotic symptoms (F31.2, 31.5, 32.3, 33.3) (International Statistical Classification of Diseases and Related HealthProblems: Tenth Revision);[18]
► Having self-reported difficulties going outside their home primarily due to anxiety that they would like treated;
► And participant is willing and able to give informed consent for participation in the trial.

## Exclusion criteria

► Unable to attempt an Oxford-Behavioural Assessment Task (O-BAT) (the primary outcome measure) at baseline (eg, due to being unpermitted to leave a psychiatric ward);
► Photosensitive epilepsy;
► Significant visual, auditory or balance impairment;
► Current receipt of another intensive psychological therapy (or about to start it within the 6 week trial therapy window);
► Insufficient comprehension of English;

► In forensic settings or psychiatric intensive care unit;
► Organic syndrome;
► Primary diagnosis of alcohol or substance disorder or personality disorder;
► Significant learning disability;
► Or current active suicidal plans.

## Assessments

Basic demographic and clinical data will be collected (eg, age, gender, ethnicity, clinical diagnosis, medication use). The primary outcome, avoidance and distress of everyday situations as measured by the O-BAT (adapted from 14), will be measured at baseline, 6 weeks and 26 weeks. The O-BAT comprises a personalised hierarchy of five real world tasks that the patient finds difficult due to anxiety. The person then tries to carry out the hierarchy, rating anxiety at each step achieved, and stopping when they decide the anxiety is too great. This therefore produces an avoidance score (0 to 5, with higher scores indicating lower avoidance) and a distress score (0 to 10, with higher scores indicating greater distress) for each level. All assessors receive training in administering the O-BAT and the manual. The initial O-BAT for each participant is reviewed by a clinician. A detailed assessment of social avoidance is first carried out using both a semi-structured interview and a self-report measure of social avoidance, the self-report O-BAT.[19] This identifies the everyday situations and tasks that are anxiety-provoking for the participant, and provides a predicted level of distress for each. Based on this, the five-step personalised hierarchy is developed. A hierarchy can be constructed within one or a number of feared situations (eg, standing on the front door step for 3 min, standing outside the front gate for 3 min, walking down the local street, walking to the local shop, buying something in the shop). The hierarchy is set up so that it is likely that the patient may only complete a small number of steps at baseline. Secondary outcomes will also be assessed at all three time-points. Anxious avoidance (AMI-A[20] and self-report version of the O-BAT[19]), suicidal ideation (Columbia Suicide Severity Rating Scale),[21] overall paranoia (R-GPTS),[22 23] paranoia worries (Paranoia Worries Questionnaire)[24] and levels of depression (PHQ-9)[25] will be assessed. Activity levels will be assessed using actigraphy (over 7 days), complemented with a time-budget assessing meaningful activity.[26] The EQ-5D-5L[27] and ReQol[28] will assess quality of life. Additionally, quality of life will be assessed using the Questionnaire about the Process of Recovery (QPR).[29] For mediation, we will assess, at all time-points, threat cognitions and use of defence behaviours (CDBQ)[30] and strength of safety, vulnerability and threat anticipation beliefs.[31] Moderators will be assessed at baseline only by a brief assessment of negative hallucinations when outside,[32] the Beck Hopelessness Scale,[33] the Body-Esteem Scale for Adolescents and Adults[34] and the Cognition and Defence Behaviours Questionnaire.[30] We will record service use, and other relevant health economic data, using the Client Service Receipt Inventory.[35] A summary of the measures is provided in table 1.

| Table 1 | Summary of objectives and assessment measures | |
|---|---|---|
| | **Objectives** | **Outcome measures** |
| Primary | Test whether the virtual reality treatment leads to reduction in avoidance and distress in everyday situations. | Oxford - behavioural assessment task (O-BAT). |
| Secondary | 1. Test clinical improvements by treatment type in activity levels, psychiatric symptoms, quality of life. | Activity levels: Actigraphy, time-budget measure. Psychiatric symptoms: Agoraphobia mobility inventory-avoidance, self-report O-BAT, Revised Green *et al* Paranoid Thoughts Scale, Paranoia Worries Questionnaire, PHQ-9, Columbia-Suicide Severity Rating Scale. Quality of life: EQ-5D-5L, ReQol, Questionnaire on the Progress of Recovery. |
| | 2. Determine the cost-effectiveness of the virtual reality treatment. | Client Service Receipt Inventory. |
| | 3. Test mediation of treatment effects by changes in safety beliefs, threat cognitions, and defence (safety-seeking) behaviours. | Cognition and Defence Behaviours Questionnaire and strength of safety beliefs, vulnerability belief and threat anticipation. |
| | 4. Test moderation of treatment effects (negative auditory hallucinations when outside, hopelessness, appearance concerns and threat cognitions). | Hallucinations scale; Beck Hopelessness Scale; Body-Esteem Scale for Adolescents and Adults; Cognition and Defence Behaviours Questionnaire. |
| | 5. Assess patient satisfaction with the VR therapy. | Modified version of the Client Satisfaction Questionnaire. |

VR, virtual reality.

## The VR psychological treatment

The gameChange VR treatment is a virtual-reality application recommended for adults (16+) who have anxieties when outside in everyday social situations. This software is intended to reduce anxieties around other people and therefore to help participants feel safer and more comfortable around people. The aim for the outcome is that patients feel more able to go outside into everyday situations. The treatment was programmed by the University of Oxford spin-out company Oxford VR (www.oxfordvr.org). The treatment is a CE marked Class I Active Medical Device- Z301 (Standalone Software), in conformity with the essential requirements and provisions of the EC Directive 93/42/EEC (Medical Devices).

A mental health professional, most likely a peer support worker or psychology assistant, will be in the room when the treatment is given. This person will help the patient put on the VR headset and start the programme. The staff member will also encourage the person to apply the learning from VR into the real world through the setting up of homework tasks to be carried out between sessions. The applications will run through the Steam software application on a laptop computer connected to a head-mounted display and accessories. All hardware is already commercially available and has not been modified for the trial. Satisfaction will be assessed after completion of the last treatment session using a modified version of the Client Satisfaction Questionnaire.[36]

The VR Cognitive Therapy (VRCT/gameChange treatment) aims for patients to test their fear expectations around other people in order to relearn safety. The treatment is not designed as exposure therapy (participants are not asked to remain in situations until anxiety reduces) but as repeated behavioural experiment tests (to learn that they are safer than they had thought). The treatment is designed to be delivered in approximately six sessions of 30 minutes. Three sessions will be considered the minimum (adherent) dose of therapy. However participants can proceed at their own pace, meaning that a fewer or greater number of sessions is allowed. The participant typically stands, and is able to walk a few paces in the scenarios. A virtual coach guides the person through the treatment, including encouraging the dropping of defence behaviours, and elicits feedback to tailor the progression of the treatment. When first entering VR, the patient goes into the coach's virtual office and is guided in how to use VR (ie, the basic functions). At the beginning of the first session, the virtual coach explains the rationale behind the treatment, and the participant selects which one of six virtual reality situations that they would like to begin in. The six virtual reality scenarios are a: café, general practitioner waiting room, pub, bus, street scene and shop. Each scenario has five levels of difficulty (eg, the number and proximity of people in the social situation increases) and participants work their way through each level of difficulty. There are (therapeutic) game type tasks within a number of the levels (that are designed to help the person drop defence behaviours and make new learning). The participant can choose a different scenario in each session or repeat a previous situation (and level). Throughout the sessions, participants' responses to questions from the virtual coach are

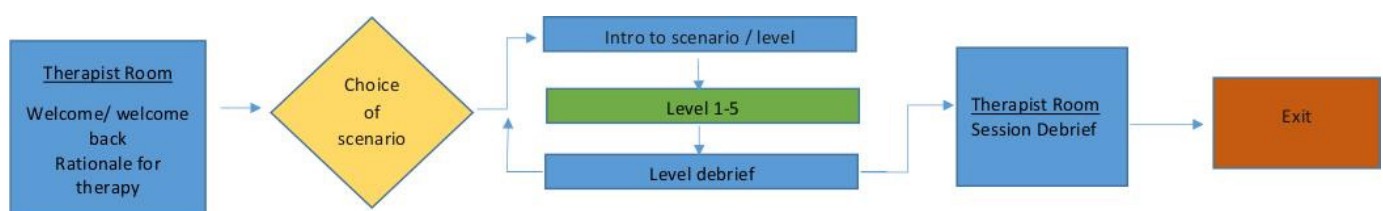

**Figure 2** The structure of the virtual reality treatment.

given by means of gripping a virtual globe. Belief ratings are repeated within VR at the beginning and end of each treatment session. Figure 2 provides a summary of the treatment design. A video about the gameChange treatment can be viewed here: https://www.youtube.com/watch?v=D31wodNAMZA.

### Control condition

Participants who are allocated to the control arm will continue to receive their usual care. No additional interventions will be offered by the research team. Treatment as usual for the participants within this trial will typically consist of long-term prescription of psychiatric medications, and meetings with a mental health practitioner. Treatment as usual will vary across individuals and mental health trusts. We will collect detailed data on treatment as usual (which will also inform the health economic evaluation).

### Adverse events

A trial standard operational procedure has been written for adverse events. We will record the occurrence of any serious adverse events reported to us and also check each patient's medical notes at the end of their participation in the trial. An adverse event is defined by the ISO14155:2011 guidelines for medical device trials as serious if it: (a) results in death or, (b) is a life-threatening illness or injury or, (c) requires hospitalisation or prolongation of existing hospitalisation or, (d) results in persistent or significant disability or incapacity or, (e) medical or surgical intervention is required to prevent any of the above, (f) leads to foetal distress, foetal death or consists of a congenital anomaly or birth defect or (g) is otherwise considered medically significant by the investigator.

Life threatening in the definition of a serious adverse event refers to an event in which the subject was at risk of death at the time of the event; it does not refer to an event that hypothetically might have caused death if it were more severe. A planned hospitalisation for a pre-existing condition, without a serious deterioration in health, is not considered to be a serious adverse event. The sorts of serious adverse events that can typically happen to this participant group include: deaths, suicide attempts, serious violent incidents and admissions to hospital.

We will also record any adverse device effects from the VR treatment, which includes adverse events resulting from insufficient or inadequate instructions for use, deployment, installation or operation, or any malfunction of the software. It also includes any event resulting from user error or intentional misuse.

### Analysis

A full statistical analysis plan will be drafted prior to recruitment beginning and approved before any analysis. We will report data in line with the Consolidated Standards of Reporting Trials 2010 Statement[37] showing attrition rates and loss to follow-up. The primary analyses will be carried out using the intention-to-treat principle. That is, after randomisation, participants will be analysed according to their allocated intervention arm irrespective of what intervention they actually receive, and with data available from all participants included in the analysis including those who do not complete therapy. The outcome analyses will be conducted by statisticians in the University of Oxford Primary Care Clinical Trials Unit.

We will test the primary hypothesis for between-group difference in the primary outcome (O-BAT at 6 weeks) using a linear mixed effects model which models the response at 6 weeks and 26 weeks, with baseline outcome measure, stratification variables and treatment assignment as fixed effects, with a patient specific random intercept. An interaction between time and randomised group will be fitted as a fixed effect to allow estimation of treatment effect at all time points. The linear mixed effects model will account for missing data assuming data are missing-at-random. Standard residual diagnostics will be assessed for the appropriateness of the model. P value <0.05 will be used as the level of statistical significance. Similar mixed effect models will be used to analyse secondary outcomes. We will recruit around 432 participants into this trial, with 216 in each arm. This sample size takes into consideration a maximum attrition rate of 20%, and provide 90% power to detect a difference of around 8 (SD=23) in O-BAT anxiety score (using the 0 to 100 scaling from Freeman *et al*, 2016[14]), from randomisation to 6 weeks (ie, standardised effect size of 0.35) at 5% level of significance (two-sided).

The mediation analysis will investigate putative mediational factors using modern causal inference methods. This involves using parametric regression models to test for mediation of VRCT on outcome through the putative mediators. Analyses will adjust for baseline measures of the mediator, outcomes and possible measured confounders. We will include repeated measurement of mediators and outcomes to account for classical measurement error and baseline confounding. The identified moderator variables (negative auditory hallucinations, hopelessness, appearance concerns and social phobia)

will be considered for moderation of the intervention effect on the primary outcome.

A microcosting approach will be used to inform the cost per patient of the VR treatment. The within-trial health economic analysis will describe and compare the costs and outcomes of the two trial arms. Incremental cost per activity gained (primary outcome) will be estimated and the costs and remaining outcomes (utilities, psychiatric symptoms and well-being) assessed separately. This will be informed by a health economics statistical plan written prior to the economic analysis. The health economics will use an NHS and social care services perspective with resource utilisation valued using national cost data sets and EQ-5D-5L data converted into utilities using the UK tariffs. A broader perspective including lost earnings, patient out-of-pocket costs and criminal justice costs will also be considered. A state-transition model will be developed to extrapolate the within-trial analysis and estimate the incremental costs per quality-adjusted life years (QALYs) gained from using the VR treatment, supported by the trial data, literature reviews and discussions with clinical experts. Uncertainty around the incremental cost-effectiveness ratio will be reported using the cost-effectiveness plane and the cost-effectiveness acceptability curve. The maximum reimbursable price of the VR treatment conditional on the willingness to pay per QALY will be determined. We will then estimate the affordability to the NHS of a decision to implement the VR treatment. This will take the form of budget impact analysis using a time horizon of 3 years to be consistent with National Institute for Health and Care Excellence (NICE), informed by the results of the trial health economics analysis.

## Patient and public involvement

The project has had extensive patient and public involvement (PPI). Principally this has occurred via The McPin Foundation, a charity that exists to 'transform mental health research by putting the lived experience of people affected by mental health problems at the heart of research methods and the research agenda'. A grant-holder is from The McPin Foundation. Three other people with lived experience commented on the grant application and a focus group of people with lived experience was convened so that they could try VR and comment on the application.

Following the award of the grant there has been considerable PPI. A LEAP has been formed to advise and shape the development of the treatment, the trial protocol and implementation into services. The LEAP comprises 10 individuals with lived experience of psychosis drawn from each of the study sites (Bristol, Manchester, Newcastle, Nottingham, Oxford). For the protocol they have advised on: the choice of outcome measures, recruitment methods, the format of recruitment materials and the content and wording of study materials. The LEAP have also reviewed and commented on the trial protocol document. In addition to the LEAP, we have also worked with people with lived experience from each of the trial sites to develop the VR treatment. A number of workshops were held. Through these workshops, people have contributed to the selection of the VR scenarios, the therapeutic tasks within the scenarios and style of VR coach. These workshops entailed people with lived experience sharing their ideas, reviewing design concepts, and testing these out within VR. In addition to these workshops, there has been weekly input from a smaller group of individuals with lived experience to gain prompt feedback on details of design. There has been detailed user testing of the VR treatment during software development.

PPI will continue throughout the trial. First, LEAP meetings will occur over the course of the trial. The LEAP will advise on any difficulties that occur in the trial. The LEAP will also contribute to the dissemination strategy. Second, there will be a qualitative evaluation of the VR treatment, with the interviews carried out by researchers with lived experience. This work will be run by The McPin Foundation. Third, a McPin staff member sits on fortnightly gameChange review meetings and on the Research Steering Committee comprised of senior team members.

## Ethics and dissemination

The trial has received Health Research Authority and Health and Care Research Wales approval (IRAS 256895, The gameChange trial). The trial received ethical approval from the NHS South Central - Oxford B Research Ethics Committee (19/SC/0075). The results of the trial will be published in a peer-reviewed journal and made open access. An anonymised version of the main outcome data will be available from the trial team on reasonable request after publication of the main results paper.

## Trial status

The trial is due to start patient recruitment in July 2019. Recruitment will be for a year until July 2020, with final outcome data collected by January 2021. A trial paper with the outcome results should be submitted for publication around April 2021.

**Author affiliations**
[1]Department of Psychiatry, University of Oxford, Oxford, UK
[2]Oxford Health NHS Foundation Trust, Oxford, UK
[3]NIHR Oxford Health Biomedical Research Centre, Oxford, UK
[4]Primary Care Clinical Trials Unit, Nuffield Department of Primary care Health Sciences, University of Oxford, Oxford, UK
[5]The McPin Foundation, London, UK
[6]National Institute of Health Research (NIHR) MindTech Med Tech Co-operative, Nottingham, UK
[7]Division of Psychiatry & Applied Psychology, School of Medicine, Institute of Mental Health, University of Nottingham, Nottingham, UK
[8]NIHR Nottingham Biomedical Research Centre, Nottingham, UK
[9]Bioengineering Reserch Group, Faculty of engineering, University of Nottingham, Nottingham, UK
[10]Nuffield Department of Population Health, University of Oxford, Oxford, UK
[11]Greater Manchester Mental Health Foundation Trust, Manchester, UK
[12]Division of Psychology and Mental Health, University of Manchester, Manchester, UK
[13]Avon and Wiltshire Mental Health Partnership (AWP) NHS Trust, Bath, UK
[14]Northumberland, Tyne and Wear NHS Foundation Trust, Newcastle upon Tyne, UK
[15]University of Newcastle, Newcastle upon Tyne, UK
[16]Nottinghamshire Healthcare NHS Foundation Trust, Nottingham, UK

[17]Department of Experimental Psychology, University of Oxford, Oxford, UK

**Correction notice** This article has been corrected since it was published. The affiliations have been updated.

**Contributors** DF is the chief investigator, conceived the project, had overall responsibility for the treatment design and the trial design and drafted the trial protocol. DMC, MC, JG, CH, TK, JM, JL, FW and L-MY contributed to the overall study design. FW, SL, TK, TLC and DMC contributed to the treatment design. TK, TLC, HA and DR have been responsible for PPI. JM, SB, MC, AB, HA, and DR have advised on treatment implementation. LM-Y is responsible for the statistical analysis. JL and MC are responsible for the health economic analysis. AR and AG contributed to the virtual reality programming and support of the hardware. SL and LR are co-ordinating the trial. The trial site leads are DF/FW (Oxford), AM (Manchester), RD (Newcastle), EO'R (Nottingham), KC (Bristol). All authors commented on the trial protocol.

**Funding** The trial is funded by the NHS National Institute for Health Research (NIHR) invention for innovation (i4i) programme (Project II-C7-0117-20001). It is also supported by the NIHR Oxford Health Biomedical Research Centre.

**Competing interests** DF is a founder and chief clinical officer of Oxford VR, a University of Oxford spin-out company, which has programmed the gameChange treatment, is a collaborator in the research, and will commercialise the treatment. DF holds equity in Oxford VR.

**Patient consent for publication** Not required.

**Provenance and peer review** Not commissioned; externally peer reviewed.

**ORCID iDs**
Daniel Freeman http://orcid.org/0000-0002-2541-2197
Felicity Waite http://orcid.org/0000-0002-2749-1386

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
