## [Reviewer comments · BMJ Open]

ARTICLE DETAILS

TITLE (PROVISIONAL)	Automated virtual reality (VR) cognitive therapy for patients with psychosis: study protocol for a single-blind parallel group randomised controlled trial (gameChange)
AUTHORS	Freeman, Daniel; Yu, Ly-Mee; Kabir, Thomas; Martin, Jen; Craven, Michael; Leal, Jose; Lambe, Sinead; Brown, Susan; Morrison, Anthony; Chapman, Kate; Dudley, Robert; O'Regan, Eileen; Rovira, Aitor; Goodsell, Andrew; Rosebrock, Laina; Bergin, Aislinn; Cryer, Tillie; Robotham, Daniel; Andleeb, Humma; Geddes, John; Hollis, Chris; Clark, David; Waite, Felicity

VERSION 1 - REVIEW

REVIEWER	Wim Veling University of Groningen, University Medical Center Groningen, the Netherlands I am involved in several VR treatment studies for people with psychotic disorders, license agreement between employer (UMCG) and Dutch VR software company (CleVR BV).
REVIEW RETURNED	17-Jun-2019

GENERAL COMMENTS	This is an exciting randomized controlled trial. The rationale of the study is clear. Many patients with psychosis do not receive psychological treatment because there is a dearth of therapists, and/or patients are too anxious to do behavioral experiments in real life. An efficacious automated brief VR treatment would be a game changer indeed. Some minor issues: - p8: a member of the clinical team should give approval for a patient to enter the trial. Is that really necessary? What if a patient wants to participate and the clinical team thinks he should not? What did the ethical committee advice on this issue?- Primary outcome: the description of the OBAT needs more detail. How is the hierarchy of five situations created? How do the researcher know that the patient really has carried out the hierarchy? What if there are less than five situations? What is the validity / reliability of this measure?- Control condition: is CBT for social anxiety or persecutory delusions allowed during the study period?- VR treatment: there still is a mental health professional present at each session. What is the exact role of this person? Is she allowed to help when the patient does not understand the VR program? Or
---

	if the patient quits after a few minutes? What does it mean that the professional "encourages" the patient?  - Presence of a staff member means that patients in the VR condition receive more attention and care from staff than in control condition. How can this effect be measured/tested? - The PI is founder and CCO of the company that will commercialize the VR treatment. What measures are taken to guarantee objectivity, e.g., independent data analysis?
--	--

VERSION 1 – AUTHOR RESPONSE

“This is an exciting randomized controlled trial. The rationale of the study is clear. Many patients with psychosis do not receive psychological treatment because there is a dearth of therapists, and/or patients are too anxious to do behavioral experiments in real life. An efficacious automated brief VR treatment would be a game changer indeed.”

We thank the reviewer for this appreciation of the trial.

“Some minor issues:

- p8: a member of the clinical team should give approval for a patient to enter the trial. Is that really necessary? What if a patient wants to participate and the clinical team thinks he should not? What did the ethical committee advice on this issue?”

We believe it is important to work as closely as possible with the responsible clinical team, in order to ensure appropriate co-ordination of care. We would not enter a patient into the clinical trial if the treating clinical team did not want this to occur. This is the basis of the ethical approval that has been received for the trial.

“- Primary outcome: the description of the OBAT needs more detail. How is the hierarchy of five situations created?”

This is a helpful comment and further detail has been added to the paper (page 9). All assessors have received training in administering the OBAT and the manual. Each patient’s initial O-BAT is reviewed by a clinician. A detailed assessment of social avoidance is carried out using both a semi structured interview and a self-report measure of social avoidance, the SR-OBAT. This identifies the everyday situations and tasks that are anxiety provoking for the participant, and provides a predicted level of distress for each. Based on this, a hierarchy of 5 steps is then constructed collaboratively with the participant. Selection of steps is guided by the predicted levels of distress so that step 1 is something this is moderately distressing, and building up to step 5, which is something extremely distressing.

“How do the researcher know that the patient really has carried out the hierarchy?”

The task has to rely on self-report. Following the OBAT, the researcher does review with the patient what occurred during the tasks. This provides a check that each step was completed as planned.

“What if there are less than five situations?”

It is five tasks rather than situations, and we have made this clearer in the text.

“What is the validity / reliability of this measure?”

The main strength of behavioural avoidance tasks is the face validity. We will be publishing a psychometric paper on the O-BAT, which is already being used in an on-going trial of ours (THRIVE). Of course behaviour avoidance tasks have long been used in the assessment of anxiety (e.g. Castagna, Davis & Lilly, 2017) and studies have reported high test-retest validity (e.g. Hamilton and King 1991; Ollendick et al. 2011) and good convergent and divergent validity and sensitivity to treatment (e.g. Steketee et al., 1996).

- "Control condition: is CBT for social anxiety or persecutory delusions allowed during the study period?"

As set out in the exclusion criteria: Current receipt of another intensive psychological therapy (or about to start it within the 6 week trial therapy window) would prevent trial entry. Therefore if the person was currently receiving such treatment, or about to start it, then they would not enter the trial. However participants could start receiving such treatment within usual care within the 6mths in the trial (though this would actually be rare in the UK and, in any case, given the exclusion criterion, would most likely happen after the post-treatment assessment). If it did occur at any time during trial participation then the data would be captured in our review of medical notes.

"- VR treatment: there still is a mental health professional present at each session. What is the exact role of this person? Is she allowed to help when the patient does not understand the VR program? Or if the patient quits after a few minutes? What does it mean that the professional "encourages" the patient?"

Thank you for allowing us to clarify this. The key role is helping set up the VR session with the patient (setting up the hardware, running the programme, helping the person put on the headset). The other key role is that the staff member will encourage learning applied to the real world by setting up a homework task between sessions, and this has been added to the paper (page 10). The person in the room will have been taught the treatment principles and be able to discuss these with the person. The staff member is also present in case there is any significant participant distress.

"- Presence of a staff member means that patients in the VR condition receive more attention and care from staff than in control condition. How can this effect be measured/tested?"

This is measured in terms of time. Undoubtedly the treatment package comprises both the VR and a staff member being present and this cannot be disentangled in terms of contribution to treatment effects in this trial. The point is that this delivery method can substantially improve access to evidence-based psychological therapy as it does not rely on a trained therapist being there.

"- The PI is founder and CCO of the company that will commercialize the VR treatment. What measures are taken to guarantee objectivity, e.g., independent data analysis?"

The trial has been registered so that the primary outcome is clear; the trial protocol is being published in this journal (and was submitted before the trial began) which states the main outcome and how this will be analysed; a detailed statistical analysis plan is already being drafted by the clinical trial unit (and will be completed before any analysis); randomisation is conducted by local sites using an online clinical trials unit system (Sortition); all data are entered by local sites on to the clinical trials unit data management system (OpenClinica), which is monitored by the CTU; and the analyses are being conducted by the clinical trials unit statisticians, who will produce the statistical analysis report (which will be submitted with the main trial paper). Therefore there are robust protections against influence of the trial findings. (It has now been explicitly added to the paper – page 12) that CTU statisticians will be carrying out the analysis.)

VERSION 2 – REVIEW

REVIEWER	Wim Veling University of Groningen, University Medical Center Groningen, the Netherlands
REVIEW RETURNED	23-Jul-2019
GENERAL COMMENTS	The authors have addressed by comments adequately. No further questions.